# COVID-19 Screening by Anti-SARS-CoV-2 Antibody Seropositivity: Clinical and Epidemiological Characteristics, Comorbidities, and Food Intake Quality

**DOI:** 10.3390/ijerph18178995

**Published:** 2021-08-26

**Authors:** Gabriela Macedo-Ojeda, José Francisco Muñoz-Valle, Patricia Yokogawa-Teraoka, Andrea Carolina Machado-Sulbarán, María Guadalupe Loza-Rojas, Atziri Citlally García-Arredondo, Rafael Tejeda-Constantini, Alejandra Natali Vega-Magaña, Guillermo González-Estevez, Mariel García-Chagollán, José Sergio Zepeda-Nuño, Jorge Hernández-Bello

**Affiliations:** 1Institute of Research in Biomedical Sciences, University Center for Health Sciences (CUCS), University of Guadalajara, Guadalajara 44100, Mexico; gabriela.macedo@cucs.udg.mx (G.M.-O.); biologiamolecular@hotmail.com (J.F.M.-V.); alejandra.vega@academicos.udg.mx (A.N.V.-M.); guillermo.gonzalezestevez@cucs.udg.mx (G.G.-E.); maye_999@hotmail.com (M.G.-C.); 2COVID-19 Health Situation Room, University of Guadalajara, Guadalajara 44100, Mexico; patyyokogawa@cucs.udg.mx; 3Department of Public Health, University Center for Health Sciences (CUCS), University of Guadalajara, Guadalajara 44100, Mexico; 4Department of Social Sciences, University Center for Health Sciences (CUCS), University of Guadalajara, Guadalajara 44100, Mexico; 5Institute for Research in Childhood and Adolescent Cancer (INICIA), University Center for Health Sciences (CUCS), University of Guadalajara, Guadalajara 44100, Mexico; andrea.machado5223@academicos.udg.mx; 6Department of Applied Clinical Nursing, University Center for Health Sciences (CUCS), University of Guadalajara, Guadalajara 44100, Mexico; guadalupe.loza@cucs.udg.mx (M.G.L.-R.); atziri.garcia@cucs.udg.mx (A.C.G.-A.); 7Institutional System of Safety, Health and Environment (SISSMA), University of Guadalajara, Guadalajara 44100, Mexico; rafael.tejeda@academicos.udg.mx; 8Emerging and Reemerging Diseases Diagnostic Laboratory (LaDEER), University Center for Health Sciences (CUCS), University of Guadalajara, Guadalajara 44100, Mexico; 9Department of Microbiology and Pathology, Pathology Laboratory, University Center for Health Sciences (CUCS), University of Guadalajara, Guadalajara 44100, Mexico; jsergio.zepeda@academicos.udg.mx

**Keywords:** COVID-19, serological testing, clinical characteristics, food intake quality

## Abstract

Developing countries have reported lower molecular diagnostic testing levels due to a lack of resources. Therefore, antibody tests represent an alternative to detect exposure to SARS-CoV-2 and analyze possible risk factors. We aimed to describe and compare the clinical-epidemiological characteristics and the quality of food intake in Mexican individuals with a positive or negative test to antibodies against SARS-CoV-2. We carried out antibody tests and applied a survey to 1799 individuals; 42% were positive, and diabetes was more prevalent in these cases (*p* < 0.01). No differences were identified in the blood type nor influenza vaccination between groups. Coughing, respiratory distress, muscle pain, joint pain, and anosmia were the most prevalent symptoms among seropositive cases (*p* < 0.0001). Food intake quality was similar in both groups, except for the most consumed type of fat (*p* = 0.006). In conclusion, this study supports the association of diabetes as a principal risk factor for SARS-CoV-2 infection in the Mexican population. The results do not support previous associations between blood group or influenza vaccination as protective factors against SARS-CoV-2 infection. However, frequent consumption of polyunsaturated fats is highlighted as a new possible associated factor with COVID-19, which more studies should corroborate as with all novel findings.

## 1. Introduction

Coronavirus disease 2019 (COVID-19) is a current pandemic disease caused by the severe acute respiratory syndrome coronavirus 2 (SARS-CoV-2) infection [1]. Considerable efforts have been made to contain this disease, however, the pandemic has continued active in many countries and has been characterized by infections with clinical manifestations of varying severity. With the global increase of COVID-19 cases, the accurate and early detection of positive cases is necessary for disease and patient management and to limit community infection and local outbreaks [2]. 

Quantitative reverse transcription-polymerase chain reaction (qRT-PCR) testing is the current gold standard for diagnosing SARS-CoV-2 infection; however, technical issues limit its utilization for large-scale screening in developing countries due to the insufficient molecular biology infrastructure. In these countries, serological immunoglobulin M (IgM)/immunoglobulin G (IgG) testing is an alternative for detecting SARS-CoV-2 exposure [3].

A recent meta-analysis aimed to assess the diagnostic accuracy of antibody tests for detecting active or previous SARS-CoV-2 infection [4]. This analysis evaluated 54 study cohorts with 8526 cases of SARS-CoV-2 infection. They reported low sensitivity for pooled results for IgG, IgM, IgA during the first week post symptom onset (<30.1%), with positivity rising in the second week (72.2%) and reaching its highest values in the third week (91.4%) [4]. The sensitivity reported by this study was mainly evaluated in hospitalized patients, so its reproducibility is unclear in cohorts of outpatients with milder and asymptomatic COVID-19, in whom the presence of lower antibody levels has been suggested [5].

The clinical relevance and use of serological testing are still an open debate; for this reason, this study aimed to describe the clinical and epidemiological characteristics of Mexican individuals attending a COVID-19 diagnostic module for a serologic test due to suspected SARS-CoV-2 infection. We also analyzed the relationship between IgG/IgM expression and the onset of clinical symptoms and some previously suggested susceptibility/protective factors for COVID-19, such as blood type [6,7,8,9] and influenza vaccination status [10,11,12,13]. Moreover, we evaluated the food intake quality among the study individuals as substantial stressors, as inadequate nutrition can lead to long-term effects affecting health and contribute to comorbidities associated with higher SARS-CoV-2 infection risk [14].

## 2. Materials and Methods

### 2.1. Design and Participants

Cross-sectional study. Mexican individuals who had a single blood sample taken to test for anti-SARS-CoV-2 anti-S and anti-N antibodies from July to November 2020 and who gave informed consent were included in this study.

The University of Guadalajara installed laboratories to carry out molecular tests and modules for serological tests to detect SARS-CoV-2 infections. It also established a 24/7 call-center for managing testing appointments. The service was provided to the general population, free of charge. An algorithm (Figure 1) was used to determine the eligibility of every subject. The criteria included five emergency criteria, 23 signs and symptoms, and 17 risk factors.

Seronegative patients to anti-SARS-CoV-2 with very high-risk symptoms were confirmed negative for virus infection by PCR (we had a 100% correlation). Randomly, 10% of patients seropositive to anti-SARS-CoV-2 were also evaluated to corroborate the diagnosis; in this case, we had a 90% correlation. 

### 2.2. The Setting of the Study, Materials, and Process

The participants attended the COVID-19 diagnostic module installed at the University of Guadalajara, Mexico, where they were invited to participate in the study. All subjects who agreed to participate signed informed consent forms. If individuals were under 18 years old (age of legal majority in Mexico), their parents or guardians signed informed consent forms, as long as the minor agreed.

Antibody tests were carried out using according COVID 19 qSARS-CoV-2 IgG/IgM kits to the test manufacturer’s instructions (Cellex, NC, USA), in a specially conditioned location, following the indications of the safety sheets for each chemical substance used, and the recommendations provided for the handling and final disposal of material and reagents indicated in the test manual. This test is a lateral flow chromatographic immunoassay approved by the Food and Drug Administration (FDA), which can detect antibodies against the “S” and “N” proteins of the SARS-CoV-2 virus. The clinical sensitivity reported of the assay was 93.8% (120/128; 95% confidence interval: 88.2–96.8%) and the clinical specificity of the assay was 96% (240/250; 95% confidence interval: 92.8–97.8%) [15].

The staff verified that the patients wore their masks correctly and that the previously trained healthcare personnel wore appropriate personal protective equipment. Once the test was done, the participants completed a survey that included two sections while waiting for their results (approximately 10 min). The first section consisted of 14 questions to identify the participant comorbidities, clinical, and epidemiological characteristics; the second section included 14 items of the Mini-Survey to Evaluate Food Intake Quality (Mini-ECCA v.2), previously validated in Mexico [16,17].

After it was completed, the participants received their test results. This result was written on the survey, which was subsequently placed in a container, where it remained for two weeks before being analyzed to avoid possible contamination due to the virus’s presence on the paper surface of the survey. 

### 2.3. Statistical Analysis

Demographic characteristics, comorbidities, risk contacts, and symptoms were compared between individuals’ negative or positive for anti-SARS-CoV-2 antibodies, using the Chi-square test, Kruskal Wallis, Fisher’s exact test, or the Student’s t-test. In addition, symptoms in positive individuals were compared according to seropositivity to anti-SARS-CoV-2 antibodies (IgM, IgG, IgM + IgG), using the same statistical tests. 

A cluster analysis was performed using Ward’s method for non-standardized variables to identify dietary patterns. Subsequently, a simple analysis of variance (ANOVA) was performed for each questionnaire item to compare the groups identified in the cluster analysis. Then, a discriminant analysis was performed to verify that the dietary patterns identified in the cluster analysis had significant differences between them. Finally, individual food intake habits and dietary patterns were compared using the Chi-square test. Statistical analyses were conducted using the Statistical Package for the Social Sciences (SPSS Statistics for Windows, Version 17.0. SPSS Inc., Chicago, IL, USA).

## 3. Results

### 3.1. Clinical and Epidemiological Characteristics of the Study Participants

There were 1799 individuals with a serology test result (positive or negative) during the study period; 42% of these suspect cases were positive for SARS-CoV-2 (anti-SARS-CoV-2 IgM, IgG, or IgM + IgG seropositivity), and 58% of them were negative (Table 1). Seropositive cases were older than those seronegative (*p* = 0.0001): median 37 (25th–75th percentiles: 28–49) years old vs. 34 years (25th–75th percentiles: 25–46), respectively. Of the 703 seropositive cases and 979 seronegative cases with information on age, the age range of the seropositive cases was led by the 21–30-year-old group (27.03%), followed by the 31–40-year-old group (26.74%). Seronegativity was more prevalent in younger participants, and seropositivity was dominant starting in the 31–40 year-old group and continuing to the over 60-year-old group. 

A higher proportion of both seropositive (58.99%) and seronegative (56.57%) individuals were women. Moreover, most of the individuals (seropositive or seronegative) reported a previous record of vaccination against influenza (>50%). Regarding comorbidities, overweight was the most prevalent in both study groups (19.84% in seropositive cases and 17.35% in those seronegative, *p* = 0.304); however, diabetes was more prevalent in seropositive cases in comparison with those seronegative (9.39% vs. 5.56%, respectively, *p* = 0.002). Seropositive cases declared more comorbidities than those seronegative (mean ± standard deviation (SD): 4.489 ± 2.409 vs. 3.810 ± 2.423, respectively, *p* = 0.008).

Blood type did not show differences between both study groups (*p* = 0.403). On the other hand, seropositive cases declared less knowledge about having contact with a patient with COVID-19 before undergoing the serological test (seropositive cases 67.20% vs. seronegative cases 73.63%, *p* = 0.003). However, seropositive cases had more days of contact with COVID-19 patients pre-test than seronegative cases (median 12 days [25th–75th percentiles: 7–17] vs. 10 days [25th–75th percentiles: 6–15], respectively, *p* = 0.0023). In both study groups, the contact with SARS-CoV-2 suspects was at home (46.56% in seropositive cases, and 47.17% in seronegative cases, *p* = 0.798, data not shown).

### 3.2. Symptoms of the Study Participants

The symptom findings in seropositive cases and seronegative cases are shown in Table 2. Seropositive cases reported a greater number of days since the onset of symptoms than seronegative cases (median 10 days [25th–75th percentiles: 7–14] vs. 8 days [25th–75th percentiles: 5–11], respectively, *p* < 0.0001). Similarly, seropositive cases declared a greater number of symptoms than seronegative cases (median five symptoms [25th–75th percentiles: 3–6] vs. four symptoms [25th–75th percentiles: 2–6], respectively, *p* < 0.0001). 

The most prevalent symptom in both study groups was headache (67.20% in seropositive cases and 65.96% in seronegative cases, *p* = 0.585). On the other hand, hypogeusia, diarrhea, vomiting, upset stomach, pain chest, fatigue, and dizziness were the less frequent symptoms in both study groups (prevalence ≤ 6%). There were statistically significant differences in the frequencies of coughing, respiratory distress, muscle and joint pain, and anosmia between both study groups, all of which were more frequent in seropositive cases (*p* < 0.0001).

### 3.3. Symptoms According to Seropositivity Pattern to Anti-SARS-CoV-2 Antibodies

The frequency of symptoms reported among individuals with anti-SARS-CoV-2 IgM, IgG, or IgM + IgG seropositivity was similar (Table 3). However, those who presented IgM + IgG reported cough more frequently (61.6%) than those who only presented IgM (47.5%) (*p* = 0.007). Moreover, anosmia was more frequent in those who presented IgM + IgG (61.6%) than in those who presented IgG (49.6%) (*p* = 0.034).

### 3.4. Food Intake Quality

Three dietary patterns were identified by the cluster analysis (healthy, unhealthy, and habits in need of improvement). The healthy pattern was characterized by a higher intake of water, vegetables, fish, fruit, healthy fat, oilseeds/avocado, lean meat, legumes, and a low intake of sweetened beverages, sweets, or commercially produced desserts, processed foods, and alcohol. The unhealthy pattern group frequently consumed sweetened beverages, foods not prepared at home, sweets or commercially produced desserts, processed foods, alcohol, and infrequently consumed water, vegetables, fish, and fruit. The group with habits in need of improvement frequently consumed unhealthy fat and whole grains and rarely consumed oilseeds/avocado, foods not prepared at home, and legumes. The analysis of variance showed differences in these patterns (*p* < 0.001) for each item of the questionnaire, and discriminant analysis confirmed that the dietary patterns had significant differences between them (*p* < 0.001), as was expected. 

The analysis of dietary patterns (Table 4) identified that only 34.4% of seropositive cases reported healthy food intake. In contrast, seronegative individuals presented this same pattern in 38.1% of the cases. However, this difference was not significant (*p* = 0.221).

The individual food intake habits were similar in both groups, except for the most consumed type of fat (*p* = 0.006), where 26.9% of the seronegative cases and 20% of the seropositive cases consume monounsaturated fats more frequently than other types of fats. Conversely, 69% of the seronegative individuals and 74.7% of those seropositive cases consume polyunsaturated fats more frequently. The optimal consumption of vegetables, fish, legumes, oilseeds, and avocado in both groups is infrequent (<15%). 

## 4. Discussion

Serological tests for SARS-CoV-2 are a topic of great interest for their potential to significantly enhance the diagnostic capability of healthcare services around the world in the current pandemic. However, as with all novel assays, each country should be specifically performing validating trials to understand their results’ clinical relevance [18].

A recent study showed that in COVID-19 patients, IgM and IgA antibodies to SARS-CoV-2 had been detected with a median of 5 (25th–75th percentiles: 3–6) days, while IgG was detected 14 (25th–75th percentiles: 10–18) days after symptom onset, with positive rates of 85.4%, 92.7%, and 77.9%, respectively [19]. Moreover, this study concluded that the efficiency of antibody detection by IgM enzyme-linked immunosorbent assay (ELISA) is higher than that of qPCR after 5.5 days post the onset of symptoms. In our study, overall seropositive cases showed a median time post symptom onset of 10 days with 25th–75th percentiles of 7–14 days. This range is greater than that previously reported by Li Guo et al. [19], which was taken as a cut-off point for the diagnosis algorithm applied in our study to define candidates for the serological tests. Based on this finding, we suggest that serological tests should be only recommended for patients who are at least one week past the onset of symptoms. This highlights the relevance of establishing specific diagnosis algorithms for each population because there could be different kinetics for developing of antibodies, which could be affected by the interpretation of various factors such as the test used, the severity of the disease, or even genetic factors. 

An earlier meta-analysis described no difference in the proportion of males and females infected with SARS-CoV-2 in several countries [20]. This does not appear to be the case in our study, as we found more seropositive cases in women. This result may be explained because individuals in this study were outpatients with mild or asymptomatic disease. It has been reported that the relative risk of severe COVID-19 and dying from COVID-19 is higher for men than for women in almost all age groups worldwide [21]. Therefore, this could explain the overrepresented prevalence of women in our study cohort. 

Regarding age, 27.03% of seropositive cases in this study were in the range of 21–30 years old. These results reflect those of the Centers for Disease Control and Prevention (CDC), which reported in a recent analysis that in June 2020, COVID-19 incidence showed a rapid rate of increasing prevalence, and the highest overall incidence was among young adults [22]. This data is alarming since this population includes the social sector supporting the economy in most countries, including Mexico.

COVID-19 is characterized by many signs and symptoms shared with other infectious diseases [23]. In the present study, coughing, respiratory distress, muscle and joint pain, and anosmia were the more prevalent symptoms in the seropositive cases than in the suspected cases that were seronegative. Therefore, we suggest that these symptoms should be considered major criteria in the decision-making to undergo a serological test for suspected COVID-19 and should also be considered major criteria to suggest the performance of a serological test for the diagnosis of COVID-19. This finding applies to every anti-SARS-CoV-2 antibody pattern studied (IgM, IgG, or IgM + IgG). One interesting finding is that anosmia was overrepresented in patients with the presence of IgM antibodies, thus, this symptom could be considered as a marker of active disease. This finding is consistent with the recent study by Semmler et al., where continuously higher detection rates of IgM were observed within 2-day intervals post-onset of symptoms for S1- and NCP-specific antibodies [24].

In COVID-19 patients who are symptomatic, age and the presence of one or more underlying comorbidities have been linked to increased disease severity. A study in 22,757 COVID-19 patients from different countries reported that major comorbidities in this population were hypertension (27.4%), diabetes (17.4%), cardiovascular disease (8.9%), chronic obstructive pulmonary disease (7.5%), cancer (3.5%), and chronic kidney disease (2.6%) [25]. 

Non-transmissible chronic disease is the main public health problem in Mexico; coronary heart disease and diabetes are the two of the leading causes of death in this country [26]. These population characteristics could be significant factors associated with the high COVID-19 mortality in Mexico. As of 15 May 2021, Mexico has reported the fourth-highest number of total SARS-CoV-2 related deaths worldwide [27].

In the present study, we observed high comorbidity prevalence in seropositive and seronegative cases; however, diabetes was the only comorbidity with a higher prevalence in seropositive cases compared to seronegative cases. Thus, we propose this comorbidity as the major risk factor for COVID-19 in our population; in contrast, other countries such as China, South Korea, Italy, the United States (USA), and the United Kingdom (UK) have reported hypertension as a major comorbidity [25]. This risk factor should also be analyzed in depth by the Mexican health authorities since previous studies support that this is a significant public health problem in this population and that it is a trigger for severe COVID-19, especially in people with social disadvantages and limited access to medical care [28].

Similar to the presence of comorbidities, blood type and influenza vaccination status have been suggested as risk factors for the development of COVID-19. Previous studies showed that blood group A was associated with a higher risk for acquiring COVID-19 compared with non-A blood groups, whereas blood group O was associated with a lower risk for infection and mild disease [8,9]. The present study has been unable to replicate this finding; therefore, further studies are needed to clarify this discrepancy in other countries and verify ancestry as a potential confounding variable for this association. 

Similar to the observations with respect to blood type, this study did not show that individuals vaccinated against influenza had a lower risk of acquiring COVID-19. In fact, 50.93% of the positive cases reported having been vaccinated during the fourth quarter of 2019 or more recently. This finding contrasts with results from other studies which reported influenza vaccination coverage rates that correlated negatively with SARS-CoV-2 seroprevalence [10], and that vaccination is associated with relative protection against COVID-19 [12]. Despite these discrepancies, we suggest that the high percentage of vaccination rate observed in the seropositive cases in this study could be associated with the moderate clinical manifestations that the patients presented since none of them had severe COVID-19. Therefore, we do not reject the theory that influenza vaccination could be a protective factor for severe COVID-19; this may be due to the previously proposed trained immunity mechanism [12].

Regarding the individual food intake habits, we observed a higher consumption of polyunsaturated fats (PUFAs) in seropositive cases than in those seronegative. Based on this observation, we suggest that the high consumption of PUFAS could be a risk factor for SARS-CoV-2 infection but also its immunoregulatory effects could be associated with a mild clinical course of COVID-19 as most of these seropositive individuals were outpatients without severe symptoms. To support this hypothesis, we analyzed the possible association between the type of fat consumed more frequently and the clinical features in seropositive cases (data not shown in the table). A relationship with headache (*p* = 0.030) and with muscle pain (*p* = 0.021) was identified. Headache was present in 76.2% of those with higher monounsaturated fats consumption and only in 65.5% of those with the highest consumption of PUFAs. Similarly, muscle pain was present in 64.9% of those with high monounsaturated fats consumption while it was present only in 53.1% of the group with the highest consumption of PUFAs. In addition, a difference (*p* = 0.043) was observed in the number of symptoms between the groups with the highest consumption of monounsaturated fats (4.96 symptoms) and PUFAs (4.38 symptoms). Further studies should analyze PUFAs intake among COVID-19 patients requiring admission to intensive care units and among outpatients to verify and reinforce these findings. In addition, it is necessary to analyze the amount and frequency of PUFAs consumption in order to have more accurate data.

Individuals highly consume PUFAs because of their potential health benefits in chronic diseases. However, numerous studies have shown that these compounds are immunoregulatory and immunosuppressive, and thus may increase susceptibility to infection [29,30,31]. These immunoregulatory effects have been mainly associated with its ability to induce anti-inflammatory cytokines such as IL-10 or regulatory T cells and therefore, the suppression of pro-inflammatory cytokines [29,32], which are associated with the severe course of COVID-19 [33].

PUFAs intake has been shown as beneficial against experimental infections caused by extracellular pathogens but these molecules have also been reported as harmful against infections caused by intracellular pathogens such as viruses, due to their inhibitory effects on some aspects of the cell mediated immune response [30]. On the other hand, Ramon et al. evaluated the ability of 17-hydroxy-docosahexaenoic acid (17-HDHA), a PUFA, in improving the immune response to the H1N1 influenza virus. Their results showed that 17-HDHA was able to increase the levels of antibodies [34]. This could also explain the relationship observed in our study between seroprevalence to anti-SARS-CoV-2 antibodies and the consumption of polyunsaturated fatty acids. However, we should consider that the Mini-ECCA v.2 evaluates the sources of fatty acids, and it is not its aim to measure the specific amount consumed per day, so it is not possible, to establish a clear relationship on frequent consumption of fatty acids as a risk factor or protective factor for COVID-19 with these data. It is for now, a result that together with previous similar findings generates a hypothesis that future studies will be able to verify.

A limitation of our study was that we did not determine the possible connection between dietary habits and social class in both study groups. Our dietary findings should be carefully scrutinized, but this study highlights sustained conclusions for a new branch of COVID-19 susceptibility studies. 

Another weakness of this study is that we did not follow up on the entire clinical course and dynamic antibody seroconversion in seropositive cases; therefore, it was not possible to associate some variants with severity of the disease or clinical outcomes. Additionally, we could not obtain a confirmatory test by PCR for every patient to confirm possible false negatives in seronegative cases. Regarding the seropositive cases that were confirmed by PCR, we observed a 90% correlation between both tests. This can be attributed to the non-concordant patients having more than 7 days post symptom onset, which considerably decreases the PCR sensitivity.

Further research is required to fully understand the association of consumption of polyunsaturated fats as a probable risk factor for SARS-CoV-2 infection. Specifically, a scrutiny of the most consumed type of food and its consumption amounts will be required.

## 5. Conclusions

In conclusion, due to the global emergency of COVID-19 and the insufficient infrastructure for molecular diagnosis in developing countries, there is a need to establish serological diagnostic testing to help diagnose the prevalence of COVID-19 in the population and detect individuals who were exposed to the virus. This study shows that antibody tests can be useful to detect COVID-19 patients, even in the acute phase. Also, it supports the association of diabetes as a principal risk factor for SARS-CoV-2 infection in the Mexican population. On the other hand, the results do not support nor reject previous associations between blood group or influenza vaccination as protective factors against SARS-CoV-2 infection. However, frequent consumption of polyunsaturated fats is highlighted as a new possible associated factor with COVID-19, which, as with all novel findings, should be corroborated by more studies.

## Figures and Tables

**Figure 1 ijerph-18-08995-f001:**
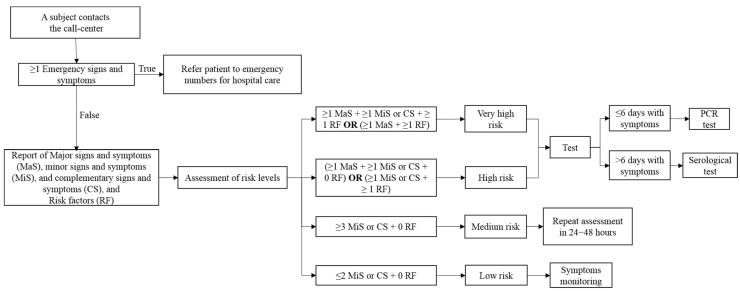
Algorithm to determine if a subject is a candidate for the test. Emergency signs and symptoms: difficulty breathing, bluish lips, chest pain, difficulty standing up, convulsions; major signs and symptoms (MaS): fever, dry cough, headache, irritability (children), loss of smell (anosmia), loss of taste (dysgeusia); minor signs and symptoms (MiS): shivers, muscular pain/soreness, bone pain, runny nose (rhinorrhea), sore throat, conjunctivitis; complementary signs and symptoms (CS): flu or cold, generally feeling unwell (malaise), tiredness/fatigue, cough with expectoration (phlegm), diarrhea, vomit, abdominal pain, fast breathing, nasal congestion; risk factors (RF): contact with suspected/confirmed COVID-19 case, healthcare worker, >60 years old, pregnancy, diabetes mellitus, hypertension, chronic obstructive pulmonary disease (COPD), asthma, immunosuppression, human immunodeficiency virus/acquired immunodeficiency syndrome (HIV/AIDS), heart disease, obesity, kidney failure, smoking, cancer, liver failure, the time frame between symptoms appearance and seeking medical care is >5 days.

**Table 1 ijerph-18-08995-t001:** Demographic characteristics, comorbidities, and risk contacts of individuals negative or positive to anti-SARS-CoV-2 antibodies.

Variables	Positive for Anti-SARS-CoV-2 Antibodies *	Negative for Anti-SARS-CoV-2 Antibodies	*p*-Value ^a^
	*n*= 756	*n* = 1043	
Age (years) ^Median (25th–75th percentiles)^	37 (28–49)	34 (25–46)	0.0001 ^b^
0–20	40 (5.69)	89 (9.09)	0.002
21–30	190 (27.03)	325 (33.20)	
31–40	188 (26.74)	236 (24.11)	
41–50	126 (17.92)	153 (15.63)	
51–60	100 (14.22)	121 (12.36)	
>60	59 (8.39)	55 (5.62)	
Sex: Female/Male ^n (%)^	446 (58.99)/310 (41.01)	590(56.57)/453(43.43)	0.304
Vaccinated against influenza ^n (%)^	385 (50.93)	541 (51.87)	0.693
**Comorbidities ^n (%)^**			
Overweight	150 (19.84)	181 (17.35)	0.179
Obesity	69 (9.13)	83 (7.96)	0.379
Hypertension	87 (11.51)	109 (10.45)	0.477
Diabetes	71 (9.39)	58 (5.56)	0.002
Immunodeficiency	8 (1.06)	10 (0.96)	0.834 ^c^
Number of comorbidities	4.489 ± 2.409	3.810 ± 2.423	0.008 ^d^
**Blood group ^n (%)^**			
A (Rh+ or Rh-)	215 (28.44)	300 (28.76)	0.403
B (Rh+ or Rh-)	61 (8.07)	94 (9.01)	
AB (Rh+ or Rh-)	19 (2.51)	32 (3.07)	
O (Rh+ or Rh-)	377 (49.87)	528 (50.62)	
Unknown	84 (11.11)	89 (8.53)	
**Risk contact**			
Contact with COVID-19 patient ^n (%)^	508 (67.20)	768 (73.63)	0.003
Days of contact with COVID-19 patients pre-test ^Median (25th–75th percentiles)^	12 (7–17)	10 (6–15)	0.0023 ^b^

* Anti-SARS-CoV-2 IgM, IgG, or IgM+ IgG seropositivity; ^a^ Chi-square, ^b^ Kruskal-Wallis, ^c^ Fisher’s exact test; ^d^ t-student. Rh = Rhesus factor; n = sample size; % = percentage.

**Table 2 ijerph-18-08995-t002:** Symptoms of individuals negative or positive for anti-SARS-CoV-2 antibodies.

Variables	Positive for Anti-SARS-CoV-2 Antibodies *	Negative for Anti-SARS-CoV-2 Antibodies	*p*-Value ^a^
	*n* = 756	*n* = 1043	
Days post symptom onset ^Median (25th–75th percentiles)^	10 (7–14)	8 (5–11)	<0.0001
Number of symptoms ^Median (25th–75th percentiles)^	5 (3–6)	4 (2–6)	<0.0001
Asymptomatic ^n (%)^	47 (6.22)	142 (13.61)	<0.0001
Headache ^n (%)^	508 (67.20)	688 (65.96)	0.585
Cough ^n (%)^	437 (57.80)	481 (46.12)	<0.0001
Fever ^n (%)^	246 (32.54)	307 (29.43)	0.159
Respiratory distress ^n (%)^	194 (25.66)	176 (16.87)	<0.0001
Runny nose, nasal congestion, sneezing ^n (%)^	384 (50.79)	559 (53.60)	0.240
Muscle pain ^n (%)^	418 (55.29)	481 (46.12)	<0.0001
Joint pain ^n (%)^	285 (37.70)	306 (29.34)	<0.0001
Sore throat ^n (%)^	374 (49.47)	546 (52.35)	0.228
Anosmia ^n (%)^	453 (59.92)	305 (29.24)	<0.0001
Hypogeusia ^n (%)^	39 (5.16)	27 (2.59)	0.004
Diarrhea, vomiting, upset stomach ^n (%)^	37 (4.89)	63 (6.04)	0.295
Chest pain ^n (%)^	5 (0.66)	12 (1.15)	0.290 ^b^
Fatigue ^n (%)^	9 (1.19)	11 (1.05)	0.786 ^b^
Dizziness ^n (%)^	5 (0.66)	12 (1.15)	0.290 ^b^

* Anti-SARS-CoV-2 IgM, IgG, or IgM+ IgG seropositivity; ^a^ Chi-square, ^b^ Fisher’s exact test. n = sample size; % = percentage.

**Table 3 ijerph-18-08995-t003:** Symptoms according to seropositivity to anti-SARS-CoV-2 antibodies.

Variables	Anti-SARS-CoV-2 IgM Seropositivity	Anti-SARS-CoV-2 IgG Seropositivity	Anti-SARS-CoV-2 IgM + IgG Seropositivity	*p*-Value ^a^
Day’s post symptom onset ^Median (25th–75th percentiles)^	*n* = 69	*n* = 66	*n* = 270	
	9.00 (7.00–11.00)	10 (7.75–13.25)	10.00 (7.00–15.00)	0.086 ^b^
	*n* = 118	*n* = 125	*n* = 513	
Number of symptoms ^Median (25th–75th percentiles)^	4.50 (3.00–6.00)	4.00 (2.00–6.00)	5.00 (3.00–6.50)	0.088 ^b^
Headache ^n (%)^	82 (69.5)	81 (64.8)	345 (67.3)	0.750
Cough ^n (%)^	56 (47.5)	65 (52.0)	316 (61.6)	0.007
Fever ^n (%)^	34 (28.8)	33 (26.4)	179 (34.9)	0.122
Respiratory distress ^n (%)^	24 (20.3)	23 (18.4)	147 (28.7)	0.022
Runny nose, nasal congestion, sneezing ^n (%)^	65 (55.1)	61 (48.8)	258 (50.3)	0.576
Muscle pain ^n (%)^	69 (58.5)	66 (52.8)	283 (55.2)	0.672
Joint pain ^n (%)^	48 (40.7)	40 (32.0)	197 (38.4)	0.324
Sore throat ^n (%)^	57 (48.3)	62 (49.6)	255 (49.7)	0.969
Anosmia ^n (%)^	75 (63.6)	62 (49.6)	316 (61.6)	0.034
Hypogeusia ^n (%)^	11 (9.3)	6 (4.8)	22 (4.3)	0.082
Diarrhea, vomiting, upset stomach ^n (%)^	7 (5.9)	2 (1.6)	28 (5.5)	0.174
Chest pain ^n (%)^	1 (0.8)	1 (0.8)	3 (0.6)	0.659 ^c^
Fatigue ^n (%)^	1 (0.8)	4 (3.2)	4 (0.8)	0.092 ^c^
Dizziness ^n (%)^	1 (0.8)	1 (0.8)	3 (0.6)	0.659 ^c^

^a^ Chi-square, ^b^ Kruskal-Wallis, ^c^ Fisher’s exact test. *n* = sample size; % = percentage.

**Table 4 ijerph-18-08995-t004:** Food intake quality in individuals negative or positive to anti-SARS-CoV-2 antibodies.

Question	Answer Options	Positive for Anti-SARS-CoV-2 Antibodies * *n* = 756	Negative for Anti-SARS-CoV-2 Antibodies *n* = 1043	*p*-Value ^a^
		n (%)	n (%)	
1. Do you drink at least 1.5 L of water per day?	A. Never	16 (2.1)	23 (2.2)	0.141
	B. Sometimes	237 (31.3)	285 (27.3)	
	C. Almost always	276 (36.5)	373 (35.8)	
	D. Always	227 (30.0)	362 (34.7)	
2. Do you consume at least 200 g of cooked or raw vegetables per day?	A. Never	17 (2.2)	35 (3.4)	0.104
	B. Sometimes	376 (49.7)	466 (44.7)	
	C. Almost always	257 (34.0)	396 (38.0)	
	D. Always	106 (14.0)	146 (14.0)	
3. Do you eat fresh or frozen fish (100 g) at least one day per week?	A. Never	122 (16.1)	146 (14.0)	0.351
	B. Sometimes	452 (59.8)	641 (61.5)	
	C. Almost always	118 (15.6)	181 (17.4)	
	D. Always	64 (8.5)	75 (7.2)	
4. Do you consume one or more glass (can) of sweetened beverages per week?	A. Never	116 (15.3)	193 (18.5)	0.319
	B. 1 to 3 times	433 (57.3)	570 (54.7)	
	C. 4 to 6 times	114 (15.1)	162 (15.5)	
	D. Daily	93 (12.3)	118 (11.3)	
5. Do you consume at least 200 g of fruit per day?	A. Never	15 (2.0)	23 (2.2)	0.880
	B. Sometimes	286 (37.8)	405 (38.8)	
	C. Almost always	294 (38.9)	408 (39.1)	
	D. Always	161 (21.3)	207 (19.1)	
6. What type of fat do you consume most frequently on a weekly basis?	A. Monounsaturated	151 (20.0)	281 (26.9)	0.006
	B. Polyunsaturated	565 (74.7)	720 (69.0)	
	C. Saturated	23 (3.0)	23 (2.2)	
	D. Do not know	17 (2.2)	19 (1.8)	
7. Do you consume at least 30 g of oilseeds or 1/2 of an avocado per day?	A. Never	54 (7.1)	61 (5.8)	0.351
	B. Sometimes	423 (56.0)	566 (54.3)	
	C. Almost always	218 (28.8)	312 (29.9)	
	D. Always	61 (8.1)	104 (10.0)	
8. Do you eat foods not prepared at home three or more days per week?	A. Never	71 (9.4)	118 (11.3)	0.576
	B. Sometimes	465 (61.5)	624 (59.8)	
	C. Almost always	152 (20.1)	202 (19.4)	
	D. Always	68 (9.0)	99 (9.5)	
9. What type of meat do you consume most often?	A. Red meat	399 (52.8)	516 (49.5)	0.361
	B. Chicken	315 (41.7)	469 (45.0)	
	C. Fish	42 (5.6)	58 (5.6)	
10. Do you eat processed foods two or more days per week?	A. Never	112 (14.8)	129 (12.4)	0.398
	B. Sometimes	567 (75.0)	792 (75.9)	
	C. Almost always	70 (9.3)	112 (10.7)	
	D. Always	7 (0.9)	10 (1.0)	
11. Do you consume sweets or commercially produced desserts two or more days per week?	A. Never	64 (8.5)	105 (10.1)	0.470
	B. Sometimes	542 (71.7)	719 (68.9)	
	C. Almost always	120 (15.9)	182 (17.4)	
	D. Always	30 (4.0)	37 (3.5)	
12. Do you eat legumes at least three days per week (300 g per week)?	A. Never	20 (2.6)	37 (3.5)	0.554
	B. Sometimes	344 (45.5)	455 (43.6)	
	C. Almost always	295 (39.0)	426 (40.8)	
	D. Always	97 (12.8)	125 (12.9)	
13. What type of cereals do you consume most often?	A. Whole grain	260 (34.4)	406 (38.9)	0.145
	B. Minimally processed	256 (33.9)	329 (31.5)	
	C. Processed and ultra-processed	240 (31.7)	308 (29.5)	
14. If you are a man, do you consume more than 2 alcoholic beverages per day? If you are a woman, do you consume more than 1 alcoholic beverage per day?	A. Never	519 (68.7)	716 (68.6)	0.077
	B. Sometimes	228 (30.2)	296 (28.4)	
	C. Almost always	8 (1.1)	26 (2.5)	
	D. Always	1 (0.1)	5 (0.5)	
Dietary patterns	Healthy food intake	260 (34.4)	397 (38.1)	0.221
	Habits in need of improvement	357 (47.2)	453 (43.4)	
	Unhealthy food intake	139 (18.4)	193 (18.5)	

* Anti-SARS-CoV-2 IgM, IgG, or IgM+ IgG seropositivity. ^a^ Chi-square.

## Data Availability

The datasets used and/or analyzed during the current study are available from the corresponding author upon reasonable request.

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
