# Peer review of "COVID-19 Screening by Anti-SARS-CoV-2 Antibody Seropositivity: Clinical and Epidemiological Characteristics, Comorbidities, and Food Intake Quality"

_ijerph, 2021, doi:10.3390/ijerph18178995_

Round 1
Reviewer 1 Report
General comments:
The authors demonstrated the usefulness of serology test for anti-SARS-CoV-2 antibody and the clinical features of seropositivity patterns in COVID-19 screening. Their article is likely to help readers to learn this field. According to their results as follows; #1. Anosmia was overrepresented in seropositivity patterns. #2. Diabetes was the only comorbidity more prevalent in seropositive cases. #3. The higher consumption of polyunsaturated fats (PUFAs) was associated with seropositive cases, although influences of polyunsaturated fats on immunoregulation against infection remain controversial. These findings let readers reconsider this field. Despite no description of new insights in this field, the review for each section has been adequately addressed in the present manuscript. Although the review for each section has been adequately addressed, several changes are required to update the manuscript.
Major:
#1. According to the authors’ results, the higher consumption of PUFAs was associated with seropositive cases. Since several evidences have suggested that PUFAs caused anti-inflammatory mechanisms against experimental viral infection, the present results may be interested and controversial to findings of previous evidences. There may be raised a possibility that immunoregulation against anti-SARS-CoV-2 infection was potentiated by frequent consumption of PUFAs in the present cohort. If so, was there any difference in clinical features or findings between frequent and infrequent consumption of PUFAs among the present cohort?
Minor:
#1. Several abbreviations of terms should be required throughout the text.
Author Response
Reviewer 1
General comments:
The authors demonstrated the usefulness of serology test for anti-SARS-CoV-2 antibody and the clinical features of seropositivity patterns in COVID-19 screening. Their article is likely to help readers to learn this field. According to their results as follows; #1. Anosmia was overrepresented in seropositivity patterns. #2. Diabetes was the only comorbidity more prevalent in seropositive cases. #3. The higher consumption of polyunsaturated fats (PUFAs) was associated with seropositive cases, although influences of polyunsaturated fats on immunoregulation against infection remain controversial. These findings let readers reconsider this field. Despite no description of new insights in this field, the review for each section has been adequately addressed in the present manuscript. Although the review for each section has been adequately addressed, several changes are required to update the manuscript.
Major:
#1. According to the authors’ results, the higher consumption of PUFAs was associated with seropositive cases. Since several evidences have suggested that PUFAs caused anti-inflammatory mechanisms against experimental viral infection, the present results may be interested and controversial to findings of previous evidences. There may be raised a possibility that immunoregulation against anti-SARS-CoV-2 infection was potentiated by frequent consumption of PUFAs in the present cohort. If so, was there any difference in clinical features or findings between frequent and infrequent consumption of PUFAs among the present cohort?
Answer: We appreciate the reviewer's insightful observation. We have performed the suggested analyses and some possible associations were found. We have reinterpreted the data in this regard, and a broader discussion was added (lines 337-354; lines 359-362). Furthermore, we add an additional limitation to our study (371-377). The conclusions were also adjusted (changes highlighted in yellow).
Minor:
#1. Several abbreviations of terms should be required throughout the text.
Answer: We reviewed the full-text document and added the description of all the abbreviations found.
Reviewer 2 Report
The manuscript from Macedo-Orjeda et al. describes differences between anti-SARS-CoV-2 antibody seropositive and seronegative individuals.
After going through the manuscript several times I came to the conclusion that the paper supports the theory that older age and diabetes might contribute to a higher risk to get COVID-19.
However, from the data presented it cannot be concluded that food intake plays an important or even weak part in the infection process.
Importantly, the vaccinated status for influenza seems to play no role in Covid risk in that study.
Major points
Reviewer: The data in table 4 did not provide sufficient evidence for any kind of food related effect on SARS-Cov-2 infection. The error rate of such an opinion poll will be definitely higher than the differences of 6% between antibody positives and negatives described for fat consumption. Moreover, p-values do not give adequate information on the accuracy of such a poll.
Minor points:
-- Please add explanations for the CI and IQR abbreviations.
-- In table 1 and 2 the asterisk (*) for antibody positivity is missing in the top row
-- Please explain once the meaning for P25th-P75th in tables 1, 2 and 3
-- In table 1, 2 and 3 percentage is given in brackets. n, % in superscript should be n (%)
Author Response
Reviewer 2
The manuscript from Macedo-Orjeda et al. describes differences between anti-SARS-CoV-2 antibody seropositive and seronegative individuals. After going through the manuscript several times I came to the conclusion that the paper supports the theory that older age and diabetes might contribute to a higher risk to get COVID-19.
However, from the data presented it cannot be concluded that food intake plays an important or even weak part in the infection process.
Importantly, the vaccinated status for influenza seems to play no role in Covid risk in that study.
Major points
Reviewer: The data in table 4 did not provide sufficient evidence for any kind of food related effect on SARS-Cov-2 infection. The error rate of such an opinion poll will be definitely higher than the differences of 6% between antibody positives and negatives described for fat consumption. Moreover, p-values do not give adequate information on the accuracy of such a poll.
Answer: The food intake quality questionnaire (Mini-ECCA v.2) was validated in two aspects, to evaluate its ability to identify dietary patterns and evaluate the reproducibility of each of its 14 questions. The results of its validation were adequate, and moderate to excellent agreement was observed in the questionnaire items. The question on the type of fat consumed most frequently showed excellent agreement (κ = 0.662, 0.654–0.671) (Bernal-Orozco, 2020). However, we should consider that the Mini-ECCA v.2 evaluates the sources of fatty acids, and it is not its aim to measure the specific amount consumed per day, so it is not possible, with these data, to establish a clear relationship on frequent consumption of fatty acids as a risk factor for COVID-19. Therefore, this finding, together with previous similar findings, generates a hypothesis that future studies will be able to verify.
We have performed new analyses to support this finding and its interpretation. We have reinterpreted the data in this regard, and a broader discussion was added (lines 337-354; lines 359-362). Furthermore, we add an additional limitation to our study (371-377).
Furthermore, we adjusted the conclusion by mentioning that frequent consumption of polyunsaturated fatty acids is a possible factor associated with COVID-19 that should be analyzed in future studies (All changes were highlighted in yellow).
Minor points:
-- Please add explanations for the CI and IQR abbreviations.
Answer: Thanks, this suggestion was made (the change was made in yellow).
-- In table 1 and 2 the asterisk (*) for antibody positivity is missing in the top row
Answer: Thanks, this suggestion was made (the change was made in yellow).
-- Please explain once the meaning for P25th-P75th in tables 1, 2 and 3
Answer: Thanks, this suggestion was made (the change was made in yellow).
-- In table 1, 2 and 3 percentage is given in brackets. n, % in superscript should be n (%)
Answer: Thanks, this suggestion was made (the change was made in yellow).
Additionally, the language and style of English were reviewed by a native of this language.
Round 2
Reviewer 2 Report
The authors have satisfactorily responded to all my questions and made the necessary changes to the manuscript.